# Antibacterial Effect of Acoustic Cavitation Promoted by Mesoporous Silicon Nanoparticles

**DOI:** 10.3390/ijms24021065

**Published:** 2023-01-05

**Authors:** Andrey Sviridov, Svetlana Mazina, Anna Ostapenko, Alexander Nikolaev, Victor Timoshenko

**Affiliations:** 1Faculty of Physics, Lomonosov Moscow State University, Leninskie Gory 1-2, 119991 Moscow, Russia; 2Research and Technical Centre of Radiation-Chemical Safety and Hygiene, FMBA, Schukinskaya St 40, 123182 Moscow, Russia; 3Faculty of Land and Environmental Management, State University of Land Use Planning, Kazakov St. 15, 105064 Moscow, Russia; 4Faculty of Ecology, Peoples Friendship University of Russia, Miklukho-Maklaya St. 6, 123182 Moscow, Russia; 5Faculty of Chemistry, Lomonosov Moscow State University, Leninskie Gory 1-3, 119991 Moscow, Russia; 6Phys-Bio Institute, National Research Nuclear University (MEPhI), Kashirskoye Sh. 31, 115409 Moscow, Russia

**Keywords:** porous silicon, silicon nanoparticles, acoustic cavitation, low-intensity ultrasound, subharmonics, antibacterial effect

## Abstract

As-prepared mesoporous silicon nanoparticles, which were synthesized by electrochemical etching of crystalline silicon wafers followed by high-energy milling in water, were explored as a sonosensitizer in aqueous media under irradiation with low-intensity ultrasound at 0.88 MHz. Due to the mixed oxide-hydride coating of the nanoparticles’ surfaces, they showed both acceptable colloidal stability and sonosensitization of the acoustic cavitation. The latter was directly measured and quantified as a cavitation energy index, i.e., time integral of the magnitude of ultrasound subharmonics. The index turned out to be several times greater for nanoparticle suspensions as compared to pure water, and it depended nonmonotonically on nanoparticle concentration. In vitro tests with *Lactobacillus casei* revealed a dramatic drop of the bacterial viability and damage of the cells after ultrasonic irradiation with intensity of about 1 W/cm^2^ in the presence of nanoparticles, which themselves are almost non-toxic at the studied concentrations of about 1 mg/mL. The experimental results prove that nanoparticle-sensitized cavitation bubbles nearby bacteria can cause bacterial lysis and death. The sonosensitizing properties of freshly prepared mesoporous silicon nanoparticles are beneficial for their application in mild antibacterial therapy and treatment of liquid media.

## 1. Introduction

Emerging bacterial pathogens are one of the most challenging threats that humanity has been facing since the 1950s and should address in the 21st century [1]. Conventional antibiotic therapy against a variety of known microbial diseases is now limited due to the multi-drug resistance among different types of bacteria induced by natural phenotypic and genetic mechanisms [2]. In this regard, the invention of new antibacterial strategies is of high priority for both science and technology. Recently, several types of inorganic (TiO_2_ [3], ZnO [4], CuO [5], Au [6], Ag [7], etc.) and organic (low molecular weight polymer-based [8], chitosan [9], intrinsically conducting polymer-based [10], etc.) nanoparticles (NPs) have been proposed as antibacterial agents. The main mechanisms lying behind the antimicrobial properties of such materials are oxidative stress caused by reactive oxygen species (ROS), strong electrostatic interaction, mechanical disruption of the membrane integrity, and photothermal effect [11,12]. Over the last decades, biocompatible [13] and biodegradable [14] porous silicon nanoparticles (PSi NPs), which can possess efficient photoluminescence at room temperature [15] and act as a photosensitizer in the generation of reactive oxygen species [16], have been extensively investigated for various biomedical applications such as luminescent bioimaging, photodynamic therapy, and targeted drug delivery (see, for example, Ref. [17]).

Ultrasound with its low invasiveness and high penetration depth is a powerful tool widely used throughout modern medicine technology [18]. It can be applied as an external trigger to activate various NPs in sonodynamic therapy (SDT) of cancer [19]. The essence of the SDT method is the combined effect of ultrasound and an active agent (sonosensitizer) which enhances the therapeutic effect [20]. Ultrasound intensities used in the SDT lie in the range of 1–10 W/cm^2^, which is several orders of the magnitude lower than for another ultrasound-assisted medical treatment known as high-intensity focused ultrasound (HIFU) [21]. Briefly, the effect of SDT is achieved based on the following processes: (i) physico-chemical destabilization of cellular structures, (ii) local hyperthermia and (iii) acoustic cavitation. The latter consists in growth and subsequent collapse of vapor-gas bubbles. The collapse is accompanied by a large local release of energy followed by the generation of free radicals reacting with various endogenous substrates to form ROS [22]. Different organic, inorganic and hybrid NPs were verified as potential sonosensitizers for the SDT [23]. Unfortunately, many types of medically applicable NPs are rather toxic [24]. As it was mentioned before, biodegradable and non-toxic PSi NPs seem to be very promising for various theranostic applications [17] and also have prospects in preclinical studies of the SDT of cancer [25].

Recently, ultrasound is also explored in the combat against pathogenic microorganisms and drug-resistant bacteria [26,27,28,29,30,31]. In the majority of studies, low-frequency or focused ultrasonic radiation of high intensity is used [26,27]. Among the serious drawbacks of such an approach are the side effects and the necessity of rather expensive equipment, which should be able to generate shock waves [27]. Several routes to synergistic antibacterial effect comprising the utilization of antibiotic drugs, microbubbles and ultrasound treatment were suggested as well [28,29]. Low-intensity medical ultrasound and SDT can also be applied as effective tools against microorganisms, including the treatment of superficial and deep-seated microbial infections [30,31]. In most cases here, the therapeutic effect is achieved due to the chemical factor of treatment related to the ultrasound-stimulated generation of ROS [31]. Recently, enhancement of the ultrasound-induced heating [32] and cavitation effects [33] by PSi NPs, as well as high-performance inhibition of cancer cell proliferation and tumor growth under the combined action of ultrasound and PSi NPs [34], has been found. Moreover, chemically oxidized Si NPs prepared from Si nanowires grown by silver-assisted etching of crystalline silicon wafers were investigated as potential sonosensitizer for the SDT of cancer cells and killing of *Escherichia coli* under irradiation with low-intensity medical ultrasound [35]. Surprisingly, the observed suppression of bacteria viability was maximal for Si NPs covered with a biopolymer (dextran) that complicated the understanding of the role of Si NPs, their surface composition, and molecular coating. Furthermore, the above-mentioned work did not investigate and clarify the possible impacts of ultrasound-induced heating and cavitation. The latter processes were found to be strongly enhanced in the presence of NPs [32,33,36,37].

The aim of our present research is to elucidate influence of acoustic cavitation promoted by mesoporous silicon nanoparticles (m-PSi NPs) with well-controlled surface coating on the viability of *Lactobacillus case* bacteria, which are chosen as well-known and widely distributed bacteria with low sensitivity to changes in the acidity level of solutions, and to understand main processes underlying the observed phenomena for potential application in antibacterial treatment. For these purposes, we recorded the dynamics of f/2 subharmonic from the spectrum of ultrasonic signal as a measure of cavitation intensity and visualize bacteria before and after the treatment with ultrasound in the presence of m-PSi NPs. Special focus was made on a possible role of the surface composition of m-PSi NPs as a key factor ruling their sonosensitizing properties.

## 2. Results and Discussion

### 2.1. Characterization of m-PSi NPs

Typical transmission electron microscopy (TEM) images of m-PSi NPs are shown in Figure 1a,b. The nanoparticles look like grains of irregular shape with sizes, i.e., maximal longitudinal dimensions, ranging from 50 to 500 nm. One could see large agglomerates in TEM images of m-PSi NPs dried on a substrate, however, in aqueous suspensions, water stands as a surfactant preventing strong agglomeration and providing their short-term colloidal stability [33] (see image of a suspension in Appendix A). Our detailed studies of the size distribution of m-PSi NPs by a scanning electron microscope (SEM) followed with an analysis of the obtained SEM images by using neural network algorithms [38] and by atomic force microscopy [39] revealed an average size of the NPs of 80 ± 25 nm. Note, there is a significant fraction of NPs with sizes varying from 50 to 100 nm, which, according to the previous studies, can easily interact with cell membrane via the electrostatic interaction [40] and endocytosis [41,42] mechanisms.

The sizes of m-PSi NPs estimated by TEM are in suitable accordance with the DLS hydrodynamic diameter distribution given in Figure 1c. The as-prepared suspension of m-PSi NPs exhibits the most probable hydrodynamic diameter at ~100 nm (black line). No significant agglomeration of as-prepared m-PSi NPs was observed for one-day storage in aqueous media at room temperature. After several days of storage, a fraction of medium and huge particle confluences appears (see broadened red and blue lines with double peaks > 200 nm in Figure 1c). The short-term colloidal stability of m-PSi NPs’ suspensions is also confirmed by the measured zeta potential value of −20 mV, which is typical for PSi NPs with a hydrophilic oxidized outer surface at neutral pH [38,43]. Despite absolute values of zeta potential greater than 25 mV for several types of suspensions at room temperature meet the criterion of incipient stability [44]:(1)zieφkT>1,
where $z_{i}$ is a charge number of ions, *e* is the electron charge, *φ* is a double-layer (zeta) potential, *k* is the Boltzmann constant, and *T* is the absolute temperature. It should be noticed that nonhomogeneously distributed local charges on the surface of NPs may induce aggregation even for high levels of zeta potential [45]. However, in the case of m-PSi NPs, which obtain a density at least 2 times lower than the density of bulk silicon (2.33 mg/cm^3^), the effective density of the material comparable to that for water can mitigate the influence of gravitation force and thus stabilizes the NPs in water. Moreover, the hydrogen-terminated surface of freshly prepared PSi [46,47] could form an “air-cushion” around its particles [48], delivering their buoyancy.

A TEM image of high resolution depicted in Figure 1b shows that m-PSi NPs consist of individual 5–20 nm sized nanocrystals. Our previous studies of similar m-PSi NPs by using a method of the isotherms of adsorption and desorption of molecular nitrogen reveled pore sizes distributed mainly from 2 to 10 nm [38]. Such pore sizes are typical for mesoporous materials [46] and are beneficial for drug loading [15,17,39], as well as for the preservation of air bubbles standing as nucleation centers of acoustic cavitation [33].

Inset in Figure 1a is an electron diffraction pattern obtained from individual m-PSi NP in the transmission mode. The periodic order of reflexes in the pattern points to the crystalline structure of silicon remained after the processes of etching and ball milling [49]. Panoramic XRD patterns for crystalline silicon (c-Si) and m-PSi NPs powders given in Appendix A demonstrate a low fraction of amorphous SiO_2_ phase in m-PSi NPs (measured as a ratio between background signal and peak intensity) and preservation of silicon crystalline structure [50]. The broadening of peaks is attributed to the presence of a significant number of small individual crystallites, the size of which can be evaluated using the Scherrer equation:(2)d=KλΔθcosθ,
where λ is the wavelength of X-ray radiation (λ = 0.154184 nm for our case), *K* is a dimensionless shape factor close to unity, Δ*θ* is an FWHM line broadening, and *θ* is a Bragg angle of the corresponding crystallographic plane. Equation (2) allows us to estimate the value of d = 10–12 nm, which agrees with early published results for the m-PSi [50].

Figure 1d shows Fourier-transform infrared (FTIR) transmittance spectra of a m-PSi film and dried m-PSi NPs. The revealed key chemical surface groups with the corresponding wavenumbers of their IR-absorption bands are summarized in Table 1. The film spectrum (black line) contains bands corresponding to various surface vibrations, the most pronounced of which are: Si-H_x_ (x = 1,2,3) stretching modes at 2070–2170 cm^−1^ as part of atomic groups such as SiH_2_-SiH, Si_2_H-SiH, Si_3_-SiH; Si-H_2_ scissoring mode at 908 cm^−1^; Si-H wagging mode at 626 cm^−1^; Si-H_2_ rolling mode at 662 cm^−1^ [51]. Such spectrum characterizes the freshly prepared PSi as hydrogen-terminated due to its passivation by HF during the process of electrochemical etching [47]. Note, weak absorption in the region of 1100 cm^−1^ corresponding to symmetric and antisymmetric stretching Si-O-Si vibrations indicates moderate oxidation of the film in air. C-H stretching bonds in the vicinity of 2900 cm^−1^ are traces of organic contamination in the resulting material received during the synthesis cycle. Low-intensity fringes across the spectrum are associated with light interference in the thin film.

The surface of freshly prepared m-PSi NPs is significantly more oxidized as compared to the initial film, which is evidenced by the rise of absorbance band magnitude in the range of 1050–1200 cm^−1^ corresponding to Si-O-Si stretching vibrations (blue line) [52]. The absorption by silicon-hydrogen bonds at 906 cm^−1^ and 2070–2170 cm^−1^ in the spectrum of m-PSi NPs’ powder is rather weak. However, some peaks from Si-H_x_ bonds are indicators of partial hydrogenation [47,53], i.e., some hydrophobicity of m-PSi NPs’ surface and pores. The preparation procedure of NPs by ball milling of m-PSi films in water results in hydrogen losses from the surface of porous silicon that eventually endows m-PSi NPs with hydrophilic properties in aqueous medium. Intensive bands corresponding to the IR absorption by Si-O-Si bonds observed in the red line of Figure 1d are evidence of strong oxidation of m-PSi NPs aged in water for several weeks. The broadband near 3500 cm^−1^ is attributed to OH stretching vibrations in residual physisorbed or chemosorbed H_2_O molecules [54].

### 2.2. Cavitation Activity in Suspensions of m-PSi NPs

The schematic of the acoustic setup built up for the cavitation activity measurement is given in Figure 2a and described in detail in Section 3.2. The photographic image of the setup is shown in Appendix A.

It is well known that inertial cavitation is accompanied by a nonlinear motion of collapsing air bubbles in the acoustic field, the signal from which has a characteristic spectrum: besides the fundamental harmonic, it obtains overtones, subharmonics, ultraharmonics, and white noise [55]. The f/2 (half of the beam main frequency) subharmonic magnitude is one of the most popular indicators used for the assessment of cavitation activity in solutions and biological media [18,56]. In our case, f/2 was equal to 0.44 MHz, so the cavitation meter was tuned to this frequency. Examples of an acoustic signal spectrum detected by the hydrophone in the cuvette filled with an investigated sample where either no or some cavitation process takes place are shown in Appendix A. The substantial rise of subharmonic magnitude (note that the spectra are plotted in the dB units) is indicative of the developed inertial cavitation process and power threshold exceeding.

Figure 2b shows time dependences of f/2 (0.44 MHz) subharmonic magnitude for aqueous suspensions of m-PSi NPs at a concentration of 1 mg/mL and pure water under ultrasonic irradiation with intensity 2 W/cm^2^. The level of subharmonic in the as-prepared suspension (blue line) exceeds that in water (black line) all over the period of observation. This fact is presumably due to the greater number of air bubble seeds on the porous surface of m-PSi NPs and, as a result, a higher rate of collapses per time unit. In the external ultrasound field, bubbles can grow out of the particle pores and eventually reach resonance sizes (~1.5 µm at 1 MHz [57]), which, when a certain intensity threshold is overcome, results in the generation of subharmonics and subsequent bubble collapse. Such growth for nanosized air nuclei is possible due to the processes of rectified diffusion [58] and the coalescence of neighboring bubbles [59]. The process of gradual oxidation of initially hydrophobic inner pores (compare the intensities of Si-H and Si-O_x_ bands for the fresh and aged suspensions in Figure 1d) in an aqueous medium leads to the degradation of sonosensitizing properties of m-PSi NPs after several weeks of their storage in the suspension (red line in Figure 2b).

Acoustic cavitation is known to be a stochastic process, which strongly depends on the initial air concentration in bubble nuclei, as well as bubble size distribution and arrangement within the sample [18]. The largest bubbles close to the resonant size collapse first, while some time is needed for the smaller ones to grow up and may lead to a delay between switching the field on and reaching the stable level of subharmonic magnitude [60]. That is why threshold values of ultrasound intensity necessary for the induction of prominent cavitation can vary between some minimal and maximal limits when acoustic pressure is gradually increased, and this cycle is repeated several times. In this view, the square of the temporal integral of subharmonic magnitude (referred to below as the cavitation energy index) gives more reliable information about energy dissipated in the system via the cavitation channel, which can be transferred into biological damage [61]. Thus, the values of cavitation energy indices for the suspension of m-PSi NPs and water calculated based on the data presented in Figure 2b differ by a factor of 4.

Figure 2c depicts the cavitation energy index as a function of m-PSi NPs’ concentration in the suspension. Each experimental point is an average of 5 independent measurements of the subharmonic magnitude over 250 s. The magnitude-time integral of subharmonic first grows up until the concentration reaches the value of 1 mg/mL, after which it starts falling. Similar nonmonotonic concentration dependences of the cavitation energy index are observed for an ultrasound with intensities of 1.5 and 2.0 W/cm^2^. The increase in the cavitation index in comparison with pure water can be attributed to the larger probability of cavitation seeds associated with m-PSi NPs. Moreover, enhanced mobility of m-PSi NPs in solution under ultrasonic irradiation and hydrogen bonding between the surface hydroxyl groups of the NPs and water molecules can initiate the appearance of short-living clusters of NPs, which could play the role of additional cavitation seeds [62]. The distortion of bonds near the boundaries of such clusters makes the nearby regions hydrophobic. The latter could further trap gas nuclei forming large cavities, which stand as reservoirs of cavitation bubbles [60]. Moreover, any change in cluster configuration under ultrasound irradiation requires additional energy making a structural contribution to the overall absorption coefficient. The described mechanism leads to the increase in the cavitation energy index; however, cavitation thresholds in the described systems could be either higher or lower than such in water. The presence of a peak value in the experimental curve can be explained by the local increase in solution viscosity, preventing its disintegration in the developed acoustic fields. This suppresses the cavitation, as well as decreases the acoustic resistance and power injected into the system.

Figure 2d shows the dependence of the cavitation energy index on the ultrasound intensity for the suspensions of m-PSi NPs of different concentrations and distilled water. These curves are the results of a single cycle of gradually lifted intensity. In this case, the threshold value of intensity sufficient for the acoustic cavitation inception can be estimated by a sharp rise of subharmonic magnitude, which corresponds to the process of bubble collapse. Differences between the cavitation thresholds in the investigated samples can be associated with the presence of submicron air nuclei in the particle pores, as well as the reduction of liquid tensile strength due to the bulk inclusions. A sharp increase followed by a steep drop in the subharmonic magnitude observed at intensity ~4 W/cm^2^ and concentration 1 g/L relates to the appearance of nonlinear modes of cavitation bubble oscillations, which are first accompanied by the general intensification of white noise and then succeeded by energy transfer to other subharmonics and ultraharmonics. This effect was numerically simulated using the Gilmore model for bubble dynamics in our previous work [33]. The discussed sonosensitizing properties of freshly prepared m-PSi NPs with no additional surface modification could compete with other more intricately synthesized types of sonosensitizers [23,33].

### 2.3. Antibacterial Effect of Ultrasound in the Presence of m-PSi NPs

Figure 3a shows the percentage of bacteria that survived after the ultrasound treatment in the presence of m-PSi NPs at different concentrations vs. time of exposure. Ultrasound intensity of 1.5 W/cm^2^ was chosen in order to exclude the undesirable heating of the irradiated area within the 20 min interval of exposure [32], which is a factor directly influencing the proliferation rate of bacteria. At the same time, this level of intensity is sufficient for the inception of cavitation activity in aqueous media (see Figure 2d). As one could expect, the sonosensitizing activity of m-PSi NPs grows with the concentration, which can be explained by the rise in the number of cavitation bubbles collapsing per time unit. Moreover, the difference in viabilities reaches its peak value for a duration between 5 and 10 min. This nonlinearity can be associated with the inhomogeneity of cavitation centers distributed within the volume of suspension, as well as the saturation of the bacterial proliferation rate decrease. Note that m-PSi NPs themselves do not influence the viability of bacteria. This fact was verified by the cell viability assay given in Appendix A: after 24 h of incubating *Lactobacillus casei* with m-PSi NPs at concentrations from 0.1 to 3 g/L, there was no substantial difference in the number of living bacterial cells relative to the corresponding number in the reference sample, while the initial numbers of colony-forming units (CFU) kept nearly the same (approximately 10 times lower than the final values). This result agrees well with the previously obtained data on low intrinsic cytotoxicity of m-PSi NPs with respect to healthy and cancer cells [43]. Wide opportunities for tuning the rate of m-PSi NPs’ degradation to non-toxic silicic acid by surface decoration [63] open the perspectives of their usage for the therapy of septic wounds.

Figure 3b represents typical electron micrographs of bacteria having an average size of ~10 µm. It is evident that the surface of bacteria is modified after their incubation with m-PSi NPs (see grains of size ~100–200 nm on the bacterial surface in Figure 3(b3)). The irradiation of bacteria with ultrasound both in the absence (Figure 3(b2)) and in the presence (Figure 3(b4–b6) of m-PSi NPs at the concentration of 1 g/L leads to the deformation of bacterial cells, but its behavior differs for the two cases. While separate ultrasound treatment leads to the “roughness” of bacterial surface attributed to the mechanical disturbance of the membrane, m-PSi NPs initiate cavitation effects such as cumulative jets [64], which result in sonoporation [65] clearly visible by hollows across the surface. Considering a low possibility of random influence of jets from bubbles far away from the bacteria, we could suppose that the interaction of m-PSi NPs with bacteria promotes the cavitation phenomena in the vicinity of the bacterial surface. It is worth noting that the visually detected sonoporation effect is not the only factor that could lead to a decrease in CFU under ultrasonic irradiation. As was shown for human keratinocytes (see Ref. [66]), there are some threshold ultrasound doses at which the apoptosis mechanism is activated, and inflammatory markers such as Interleukin 6 are overexpressed.

## 3. Materials and Methods

### 3.1. Preparation of m-PSi NPs

Mesoporous silicon layers and nanoparticles were prepared according to a standard procedure of electrochemical etching followed by high-energy milling [46]. First, a layer of porous silicon was synthesized by electrochemical etching of p-type (100)-oriented low boron-doped wafers of electronic-grade crystalline silicon (Telecom-STV, Moscow, Russia) with the specific resistivity of 1–10 Ohm∙cm in an aqueous solution of 50% HF and C_2_H_5_OH taken in ratio 1:1 [49]. The etching process was performed for 60 min at an average current density of 50 mA/cm^2^ in a double-electrode Teflon cell, where the silicon wafer served as an anode and a platinum spiral was used as a cathode. The formed mesoporous film was lifted off from a substrate in an electropolishing mode via a short 10-fold increase in the etching current. After that, the film was washed with water and dried at 65 °C for 1 h. The porosity of m-PSi was determined by using a gravimetric method [67], and it was found to be 65 ± 5%.

Second, the dried m-PSi films were mechanically ground in an agate mortar down to powder millimeter-sized grains. Then the powder was mixed with de-ionized water (1:1 by the volume), and the mixture was ball-milled in a high-energy planetary-type ball mill (Pulverisette 7 premium line, Fritsch GmbH, Idar-Oberstein, Germany) for 30 min at 1000 r.p.m. A freshly prepared suspension of mesoporous silicon nanoparticles (m-PSi NPs) was centrifuged at 2000 r.p.m. for 2 min, and the obtained supernatant was diluted to a necessary concentration for further experiments. The concentration was determined by weighing the dried residue of a sample. The suspension was additionally homogenized by an ultrasonic bath with an operating frequency of 30 kHz. The scheme of the synthesis cycle is given in Appendix A.

### 3.2. Measurements of m-PSi NPs’ Sizes and Crystallinity

The morphological analysis of m-PSi NPs was performed by using a transmission electron microscope (TEM; LEO 912 AB Omega, Zeiss, Jena, Germany). The samples of m-PSi NPs for TEM studies were prepared by depositing drops of the aqueous nanoparticle suspension on standard carbon-coated gold grids, followed by their drying in the air for 10 min. Distribution of the hydrodynamic diameters of m-PSi NPs was obtained by means of the dynamic light scattering technique (DLS) using a Zetasizer Nano ZS (Malvern Instruments, Malvern, U.K.). The NP suspensions were diluted down to low concentrations (0.01–0.02 mg/mL, almost transparent) and equilibrated at 25 °C for 5 min before the measurements. The supplied software calculated the ratio of particles of a certain size range in a total number of particles. Zeta potential measurement based on the phenomenon of laser Doppler electrophoresis was also carried out using the above-mentioned DLS apparatus. The crystallinity and phase content of m-PSi NPs were investigated by using X-ray diffractometry (XRD) with an X-ray spectrometer (Renom, ExpertCenter Ltd., Moscow, Russia) for Cu K-α line (λ = 0.154184 nm) radiation at room temperature in air. Prior to the XRD measurement, m-PSi NPs and aqueous suspensions of polycrystalline Si powder (for comparison) were deposited on thin glass substrates ad dried in air at 50 °C for 30 min.

### 3.3. FTIR Measurement

The chemical composition of the porous silicon surface was studied by Fourier-transform infrared spectroscopy (FTIR; IFS 66 *v*/*s*, Bruker Corp., USA) in the range of 6000–400 cm^−1^ with a resolution of 2 cm^−1^. An as-prepared free-standing film of porous silicon was fixed in a holder. A powder of m-PSi NPs dried from a solution was mixed with KBr to make tablets, which were also fixed in the holder. Incident IR radiation was normal to the surface of an investigated sample. First, a background IR spectrum of the air environment in an empty sample chamber was recorded. Further, an experimental spectrum was measured at room temperature and calculated as a ratio of transmitted light intensity to the signal of background at the same wavenumber.

### 3.4. Setup for the Cavitation Activity Detection

Sonosensitizing properties of m-PSi NPs were investigated in situ by using a hand-made acoustic setup. The schematic of the acoustic setup built up for the cavitation activity measurement is given in Figure 2a. A cylindrical plastic cuvette (gray cylinder) filled with an analyzed suspension of m-PSi NPs (brown rounds) or distilled water, piezoelectric transducer (orange cylinder) with operating frequency f = 0.88 MHz, and catcher (orange trapezium) absorbing the sound transferred through the cuvette were merged into a temperature-stabilized water tank (blue). The cuvette side windows of diameter ~2 cm were sound transparent, and the volume of the cuvette was ~10 mL. The transducer with a diameter of 2 cm located 3 cm from the cuvette was driven by a built-in generator supplied with alternating voltage (left yellow rectangle) and produced a weakly diverging beam of continuous ultrasound. Degassed and distilled water was used as a coupling medium. The acoustic cavitation activity inside the cuvette was detected by a wide-band probe (hydrophone; dark gray) with a thickness of 6 mm working in the frequency range of 0.01–10 MHz. The probe was inserted in the vicinity of the cuvette center at a depth of ~5 mm. The signal registered by the probe was fed to a selective voltage meter (cavitation meter; V6-10, Radiopribor, Saint Petersburg, Russia; central, yellow rectangle), which filtered, amplified, and measured (in dB and mV) the magnitude of the signal component in the desired spectral region. The operation of the generator, as well as data acquisition and processing, were automated by self-written software installed on a laptop (right yellow rectangle). In order to ensure a high level of the repeatability of experimental data, all the samples were degassed in an ultrasonic bath for 1 h prior to the measurements.

Note, besides the detection of subharmonic magnitude, we also tried to measure the cavitation activity by the level of white noise in the spectrum of the transmitted signal [55] using a cavitation meter (Laboratory of ultrasonic processes and equipment at Belarusian State University of Informatics and Radioelectronics, Belarus) in a manner it was described in our previous study [68]. Unfortunately, the absorption of ultrasound by concentrated m-PSi NPs suspensions was strong enough to provide the appropriate level of sample-to-sample sensitivity: the measured white noise spectra were almost the same for samples with concentrations from 0.5 to 1.5 g/L. The frequency of ultrasound was additionally controlled by a frequency meter. To eliminate the influence of thermal effects in the further described in vitro experiments, the temperature in the sample cuvette was recorded with a sensitive thermocouple, and the detected heating did not exceed 2–3 °C throughout the utilized range of ultrasound intensities and particle concentrations.

### 3.5. In Vitro Studies

Lyophilized cultures of *Lactobacillus casei* (AiBi, series Lb 3.02, Soyuzsnab, Krasnogorsk, Russia) were slowly thawed before the experiment and placed for 1 h in a liquid culture medium of milk skim at 24 °C then incubated for 1 h at 37 °C and finally transferred to a fresh culture medium. To achieve a homogeneous ultrasonic field, the experiments were carried out with bacterial culture in a suspended form. The initial concentration of bacteria in a suspension was 10^3^ CFU/mL. The schematic of the experimental setup is shown in Appendix A. 5 mL of the prepared suspension was poured into a glass tube with a sound-transparent bottom. The tube was submerged in a thermostatic bath, maintaining the temperature of water at 37 °C. The investigated samples containing different concentrations of m-PSi NPs were exposed to ultrasonic radiation with an intensity of 1.5 W/cm^2^ generated by a transducer with an operating frequency of 0.88 MHz for 5, 10, or 20 min. After the irradiation, 1 mL of the suspensions was transferred to an agarose culture medium in a Petri dish via the cup-plate method and placed in a thermostatic chamber at 37 °C. The number of viable bacteria was determined by counting colonies on Petri dishes after 24 h using a luminescent microscope (Micmed-6, Lomo, Saint Petersburg, Russia). For this purpose, the bacterial cells were stained with acridine orange. Viable (colony-forming) cells provided yellow-green lighting, the intensity of which was analyzed using image-processing software.

The sonosensitizing effect of m-PSi NPs was estimated by comparing the viability of *Lactobacillus casei* without and after exposure to ultrasound, either in the presence or absence of nanoparticles. The aggregate effect was evaluated as a test-to-reference percentage ratio. For cytotoxicity estimation of m-PSi NPs, the colony of bacteria was left for 24 h to proliferate. The final concentration was approximately 1 order higher than the initial one. Sterile water was added to the control sample instead of the nanoparticle suspension.

### 3.6. Scanning Electron Microscopy of Bacteria

Besides the determination of colony-forming units, bacteria were also examined with a scanning electron microscope (SEM), providing the detection of any morphological changes in the cells. To prepare the specimens, cell suspensions were infiltrated and subjected to sublimation drying. Next, the samples were mounted on a slide. Their surface was sputtered by gold with a layer thickness of 20 nm in an ion sputtering machine (IB-3, Eiko, Tokyo, Japan). The material was viewed in a scanning electron microscope (CAM-scan, Hitachi, Hitachi City, Japan) at an accelerating voltage of 15 kV and working magnification in the range from ×10,000 to ×100,000.

## 4. Conclusions

In summary, we synthesized aqueous suspensions of m-PSi NPs via the process of electrochemical etching of crystalline silicon and subsequent high-energy ball milling of m-PSi layers in water. The pores of freshly prepared m-PSi NPs could entrap nanoscale air nuclei, which grow up to bubbles of resonant sizes when irradiated with low intensive ultrasound (frequency 0.88 MHz, intensity 1–4 W/cm^2^). A custom acoustic setup allowed us to record the dynamics of f/2 subharmonic magnitude in samples, the squared temporal integral of which is a measure of cavitation energy released in the system. Such cavitation energy index appeared to be 2–5 times higher for the suspensions of m-PSi NPs at various concentrations as compared to pure water. We verified the antibacterial effect of the combined action of ultrasound and m-PSi NPs caused by enhanced cavitation, observing a dramatic decrease in bacterial viability down to 5–30% after 20 min of irradiation. At a microscale, this effect is due to sonoporation from jets produced by collapsing air bubbles in the vicinity of bacteria membranes, which leads either to bacterial death or damage that rules out the cell division. The observed sonosensitizing property of m-PSi NPs against bacteria can be proposed for application in mild antibacterial therapy, e.g., treatment of purulent wounds and in antibacterial cleaning of different aqueous suspensions and biological liquids. Taking into account the non-toxicity and biodegradability of porous silicon, as well as the relatively simple and convenient ultrasonic technique, this approach can be a suitable alternative for other antibacterial treatments using more toxic nanoparticles and chemicals.

## Figures and Tables

**Figure 1 ijms-24-01065-f001:**
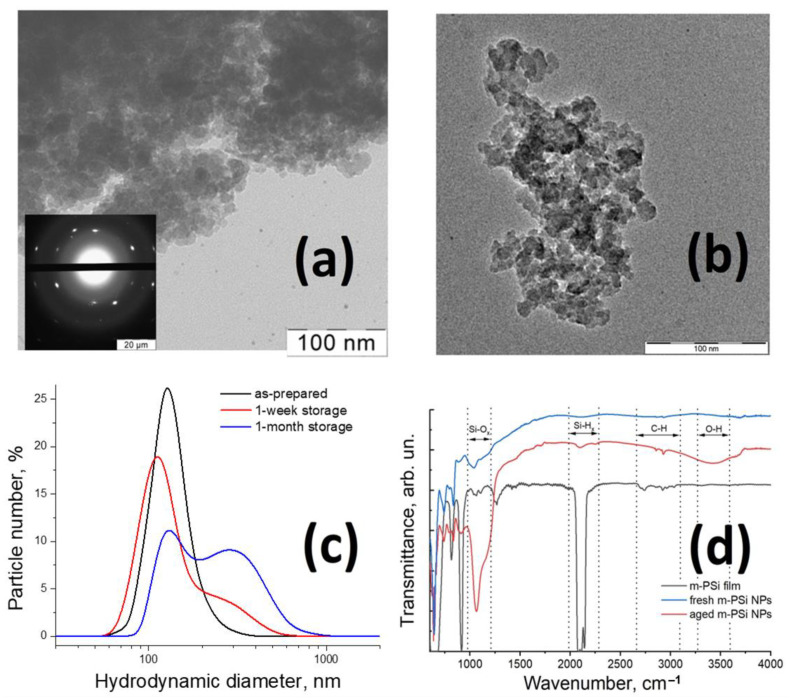
Characterization of the samples. (**a**) Typical TEM image and electron diffraction pattern of m-PSi NPs dried from suspension on a substrate. (**b**) High-resolution TEM image of a single particle. (**c**) DLS spectra of hydrodynamic diameter distribution of m-PSi NPs suspended in water just after their preparation (**black line**), after 1 week (**red line**), and 1 month (**blue line**) of storage at room temperature. (**d**) FTIR transmittance spectra of an m-PSi film (**black line**), m-PSi NPs dried from a freshly prepared suspension (**blue line**) and from a suspension aged for several weeks (**red line**). Black arrows and designations correspond to the bands of the main chemical groups.

**Figure 2 ijms-24-01065-f002:**
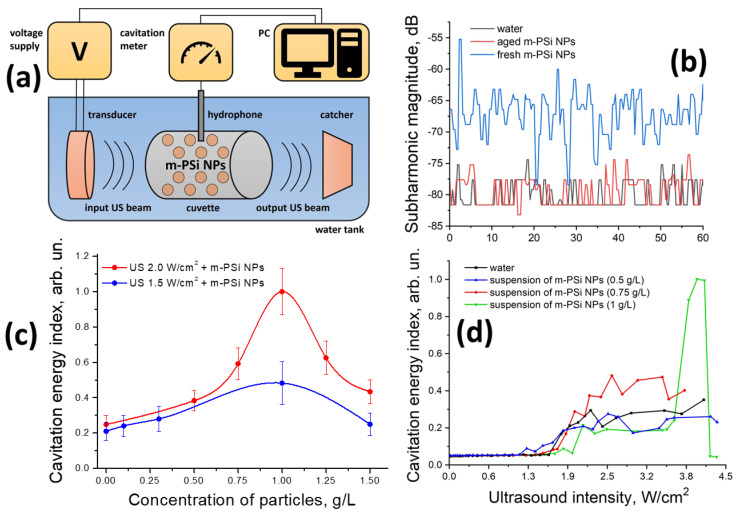
Investigation of the cavitation-sensitizing properties of m-PSi NPs: (**a**) Schematic of acoustic setup built for the cavitation activity measurement. (**b**) f/2 subharmonic magnitude in water (**blue line**), freshly prepared aqueous suspension of m-PSi NPs (**blue line**), and suspension of m-PSi NPs aged for several weeks at the concentration of 1 mg/mL (**red line**) vs. time of exposure to the ultrasound of intensity 2 W/cm^2^. (**c**) Cavitation energy index vs. concentration of m-PSi NPs in the freshly prepared suspension under exposure to the ultrasound irradiation with intensities of 1.5 W/cm^2^ (blue circles) and 2.0 W/cm^2^ (red circles). Vertical bars correspond to the RMSE, and continuous curves are Bezier splines as guides for eyes. (**d**) Cavitation energy index vs. ultrasound intensity in water (**black line**) and freshly prepared suspensions of m-PSi NPs at a concentration of 0.5 (**blue line**), 0.75 (**red line**), and 1 (**green line**) g/L.

**Figure 3 ijms-24-01065-f003:**
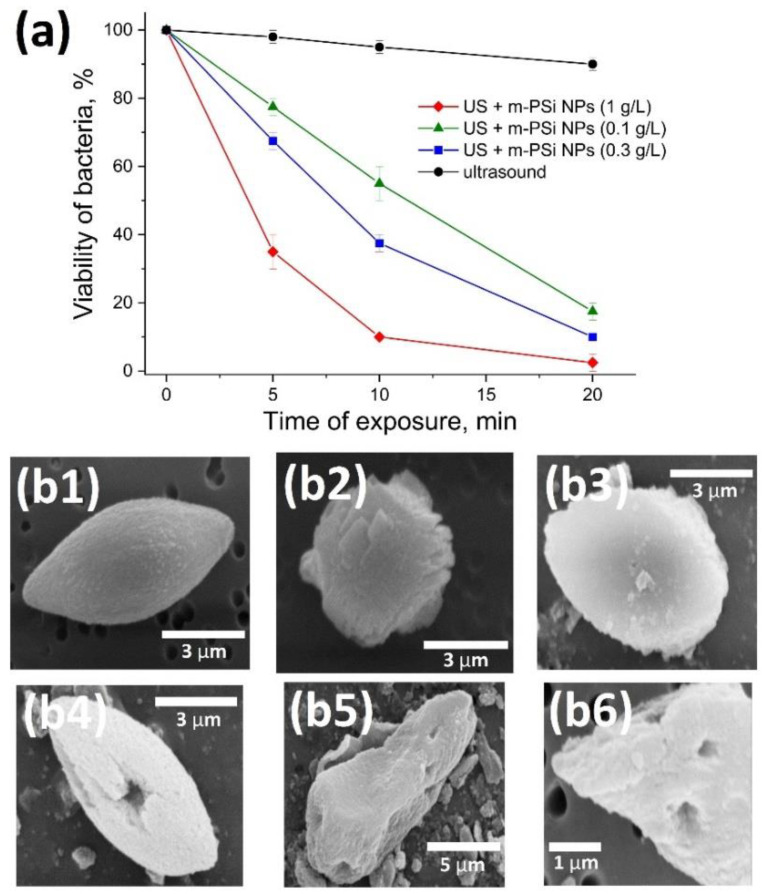
Antibacterial effect of combined action of ultrasound and m-PSi NPs. (**а**) Viability of *Lactobacillus casei* exposed to the ultrasound separately (**black line**) and in the presence of m-PSi NPs at a concentration of 0.1 (**green line**), 0.3 (**blue line**), and 1 (**red line**) g/L calculated as a number of live bacteria relative to their number in the reference colony corresponding to zero particle concentration. (**b**) Microscopic images of *Lactobacillus casei*: (**b1**) as-grown, (**b2**) after exposure to ultrasound, (**b3**) incubated with m-PSi NPs, (**b4**–**b6**) after exposure to ultrasound in the presence of m-PSi NPs at the concentration of 1 mg/mL.

**Table 1 ijms-24-01065-t001:** List of the key chemical surface groups with the corresponding wavenumbers of their IR-absorption bands [47,51].

Group	Wavenumber (cm^−1^)	Vibration Mode
Si-H	626	Wagging
Si-H_2_	662	Rolling
Si-H_2_	908	Scissoring
Si-O-Si	1080	Assym. stretching
Si-O-Si	1170	Sym. stretching
Si-H_2_	2082	Stretching
Si-H	2112	Stretching
Si-H_3_	2138	Stretching
SiO-H	3740, 3700–2700, 1640	Stretching
H_2_O	3700–2700, 1620	Stretching
C-H	2850–2950	Stretching

## Data Availability

Not applicable.

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
