# Peer review of "Antibacterial Effect of Acoustic Cavitation Promoted by Mesoporous Silicon Nanoparticles"

_ijms, 2023, doi:10.3390/ijms24021065_

Round 1

Reviewer 1 Report

Sviridov et al. synthesized aqueous suspensions of m-PSi NPs via the process of electrochemical etching of crystalline silicon and subsequent high-energy ball milling of m-PSi layers in water. They further elucidated the influence of acoustic cavitation promoted by mesoporous silicon nanoparticles (m-PSi NPs) on the viability of bacteria and understand the main processes underlying the observed phenomena for potential application in antibacterial treatment. Taking into account the non-toxicity and biodegradability of porous silicon as well as the relatively simple and convenient ultrasonic technique, this approach can be a good alternative for other antibacterial treatments using more toxic nanoparticles and chemicals. However, the following comments should be addressed before the manuscript can be published.

1. It is recommended that the authors rewrite the abstract section, where the novelty of this work needs to be highlighted.

2. What is the pore size distribution of mesoporous silica nanoparticles?

3. What is the particle size distribution of silica nanoparticles m-PSi NPs?

4. Can the authors provide cell viability assays?

5. When m-PSi NPs are used to treat septic wounds, is their leaching toxic to humans?

6. Do the authors perform biocompatibility testing of the m-PSi NPs?

7. Some grammatical errors need to be corrected.

Author Response

Comment 1: It is recommended that the authors rewrite the abstract section, where the novelty of this work needs to be highlighted.

Answer: We have rewritten the abstract to highlight the novelty of our work. We believe that now it is clear that we connect an enhancement of the directly measured cavitation intensity in aqueous media by porous silicon nanoparticles with the observed bacterial lysis and decrease of viability, which has not been done before by other researchers. Another feature is usage of freshly prepared mesoporous silicon nanoparticles with mixed oxide-hydride surface and no additional modification, which, despite the preparation simplicity, provides outstanding sonosensitizing properties.

Comment 2: What is the pore size distribution of mesoporous silica nanoparticles?

Answer: The pore sizes in studied mesoporous silicon (not silica) nanoparticles  lie in the range of 2–10 nm, that is typical for a mesoporous material. The pore size distribution in nanoparticles produced by the same method was studied in our previous paper (https://doi.org/10.1016/j.micromeso.2021.111641). We have included a brief discussion of these data to Section 2.1 (lines 131-136).

Comment 3: What is the particle size distribution of silica nanoparticles m-PSi NPs?

Answer: It is rather difficult to obtain m-PSi NPs’ size distributions based on the TEM images because of impossibility to outline individual particles in their confluence. However, this problem has been overcome by processing SEM images of m-PSi NPs using neural network algorithms, as well as visualizing m-PSi NPs with AFM on a mica substrate to prevent their agglomeration and overlapping. We included data reported in the corresponding papers (Eremina, A.S.; et. al. Mesoporous silicon nanoparticles covered with PEG molecules by mechanical grinding in aqueous suspensions. Microporous Mesoporous Mater. 2022, 331, 111641, https://doi.org/10.1016/j.micromeso.2021.111641; Konoplyannikov, M.A.; et. al. Mesoporous silicon nanoparticles loaded with salinomycin for cancer therapy applications. Microporous Mesoporous Mater. 2021, 328, 111473, https://doi.org/10.1016/j.micromeso.2021.111473). The average size of m-PSi NPs lies in the range of 50–100 nm and is in good accordance with the DLS distribution of particle hydrodynamic diameters given in Figure 1c. The corresponding data and discussion have been added to Section 2.1 (lines 91-101).

Comment 4: Can the authors provide cell viability assays?

Answer: We inserted the viability assay data in Figure S6 of the Supplementary Materials, as well as discussed the obtained low cytotoxicity of m-PSi NPs in Section 2.3 (lines 279-284).

Comment 5: When m-PSi NPs are used to treat septic wounds, is their leaching toxic to humans?

Answer: According to the literature data, m-PSi NPs during their storage in aqueous and biological media degrade to nontoxic silicic acid. The corresponding discussion has been added to Section 2.3 (lines 284-288).

Comment 6: Do the authors perform biocompatibility testing of the m-PSi NPs?

Answer: While clinical trials of m-PSi NPs on humans are just a matter of the near future, a lot of in vivo studies have been performed on mice, rats and rabbits so far Refs. [14, 15, 17, 40]. We performed a number of biological experiments both in vitro and in vivo, the results of which were described in several our papers, i.e. Refs. [34, 39, 43]. These results proved the biocompatibility of m-PSi NPs.

Comment 7: Some grammatical errors need to be corrected.

Answer:  We fully agree with this notice and revised the text to get rid of grammatical errors.

Reviewer 2 Report

The manuscript entitled "Antibacterial effect of acoustic cavitation promoted by mesoporous silicon nanoparticles" is mainly focused on investigating the influence of acoustic cavitation promoted by mesoporous silicon nanoparticles on the viability of bacteria.

The manuscript is written well and presented logically. Despite this, I have several remarks and questions about it.

The origin of the silicon used in this work is not clear to me. Was it prepared by the authors or purchased from elsewhere?

Nanoparticles are present when their size below 100 nm exceeds 50% of their content. Meanwhile, in the results presented in this paper, the conclusion is the opposite. The title of this manuscript is slightly misleading because the mesoporous silicon used in this work is not nanosized.

Is this silicon crystalline? X-ray diffraction measurement is also required.

Author Response

Comment 1: The origin of the silicon used in this work is not clear to me. Was it prepared by the authors or purchased from elsewhere?

Answer: We used commercially available electronic-grade silicon wafers purchased from a local manufacturer, and stated this fact in Section 3.1 (line 317).

Comment 2: Nanoparticles are present when their size below 100 nm exceeds 50% of their content. Meanwhile, in the results presented in this paper, the conclusion is the opposite. The title of this manuscript is slightly misleading because the mesoporous silicon used in this work is not nanosized.

Answer: We agree that the previous TEM image of m-PSi NPs did not reveal their sizes in the nanoscale.  We have changed the low-quality TEM image for a high-resolution one (new Figure 1b). Now one can see both NP agglomerates (Figure 1a) and individual NP with size about 100 nm (Figure 1b). According to our previous papers, devoted to the detailed studies of the size distribution for m-PSi NPs prepared by the same method, the mean size of those NPs is about 80 ± 25 nm. The corresponding data and discussion have been added to Section 2.1 (Figure 1b, lines 91-101).

Comment 3: Is this silicon crystalline? X-ray diffraction measurement is also required.

Answer:  Porous silicon preserves the crystalline structure of c-Si, which is evident by the XRD spectra given in new Figure 2S of Suppl. Mat. The corresponding changes were inserted into the text in Sections 2.1 and 3.2 (lines 139-147, 348-351).

Reviewer 3 Report

The manuscript reports about the preparation, characterization and use of silicon nanoparticles as sonosensitizers for antibacterial sonodynamic therapy. The topic is relevant and the experimental plan, despite a number of missing experiments, is valid. In general, the presentation of results is not clear and authors tend to force their interpretation. My specific comments are listed in the following.

Major issues

1. Authors do not contextualize adequately their research, since they never mention the existing literature about antibacterial sonodynamic therapy, whose state-of-the-art should be discussed in the introduction (see, for example, 10.1039/D2NR01847K and 10.1021/acsbiomaterials.1c00587). 

2. Based on the existing literature, the novelty of this research need to be clarified in the introduction and the new achievements should be highlighted in the conclusions.

3. In the paragraph of the introduction about the use of PSi NPs for sonodynamic therapy, authors only cite their own work (5 references!): this is inappropriate self-citation. Instead, it would be appropriate to replace most self-citations by other relevant references (as, for example, 10.1186/1556-276X-9-463). 

4. TEM and DLS techniques are described poorly: all experimental details should be reported accurately.

5. Authors should clarify what do they mean with particle size:

- in TEM, they do not specify how it is evaluated from images;

- DLS usually measures hydrodynamic radius or hydrodynamic diameter, which one of these quantities is reported in Figure 1?

- in general, authors should precisely report the specific measured quantity instead of using the generic word "size".

6. Lines 66-70. This period is wrong, since (i) no experiments about colloidal stability in time are reported in the manuscript, and (ii) the Z-potential does not guarantee the stability of a colloidal suspension, as the presence of inhomogeneous charge could induce aggregation even for high Z-potential (see, for example, 10.3390/nano12091529 and 10.1016/j.jcis.2020.07.006): modify the sentences to make them more accurate or remove them.

7. Some peaks of the FTIR spectrum that are pointed in Figure 1d are not discussed in the text (Si-O bending, C-H stretching, O-H stretching).

8. Line 95. It is not clear how the hydrophilicity of PSi NPs influences their colloidal stability: motivate this statement or remove it.

9. The quantity subharmonic magnitude should be defined to make the results comprehensible.

10. Line 119 "correlates with the more intense cavitation process in the suspension m-PSi NPs". No experiments about the intensity of the cavitation process are reported in the manuscript to substantiate this statement. Remove this sentence. 

11. Lines 120-121 "This fact can be explained by the greater number of air bubble seeds on the porous surface of m-PSi NPs". No experiments to evaluate the number of bubbles are reported in the manuscript to substantiate this statement. If it is an hypothesis, be clear, otherwise remove the sentence.

12. Figure 2c. Authors state in the text that "The smooth curve is a result of fitting": the function used for the fitting procedure, the motivation for the choice (did you fit experimental data to a theoretical model?), and the fitting parameters must be reported in the manuscript.

13. Figure 2d. The huge increase of the cavitation energy observed at 3.5 W/cm2 in the case c =1 g/L, that is absent in trends ad different concentrations, should be discussed in the text of the manuscript.

14. To substantiate the claims reported in lines 163-167, authors must report experiments (FTIR and cavitation activity) as a function of the aging of NPs.

15. In figure 3a, the following control samples are missing:

- bacteria without application of ultrasound, kept in the same conditions (transfer in the glass tube, thermostatic bath, temperature) and for the same times used in treatments;

- treatments with PSi NPs without application of ultrasound (in this respect, the sentence "m-PSi NPs themselves do not influence the viability of bacteria", reported in lines 176-177, is not supported by experiments).

16. The deformation/poration of bacterial cells observed by SEM and the (possible, control sample is missing) effects of ultrasound on the bacteria viability should be discussed in terms of ultrasound-induced mechanical and biological damage and compared to other studies performed on this phenomena (see, for example, 10.1038/s41598-017-16708-4)

17. Viability results at the same NPs concentrations used in cavitation experiments must be reported in Figure 3 and, viceversa, cavitation results at the same concentrations used in viability experiments should be reported in Figure 2.

18. Why did authors performed viability experiments at 1.5 W/cm2 instead of an higher intensity, above the cavitation threshold?

Minor comments

- In the first sentence of the Introduction, authors only list inorganic NPs, why?

- In the same sentence, authors only cite one review article published in 2012: a number of more up-to-date papers have been published meanwhile and should be cited.

- Line 62 "particles with sizes from 30 to 100 nm, which can easily attach or/and penetrate through the cell membrane". Provide references in support of this statement.

- Line 63 "Water stands as a surfactant preventing the formation of large agglomerates". In TEM experiments NPs have been dried and water is not present, therefore the reason for reporting this sentence is not clear. Please, clarify this in the manuscript or remove the sentence.

- The information given by electron diffraction pattern is not useful for the scope of the manuscript, this result could be removed.

- Figure 1c. Did you really pretend derive a (hydrodynamic radius or diameter?) distribution from DLS measurements, based on 30 counts? Please double check the presentation of this result.

- Figure 1d. This plot is a messy. (i) Please reorganize it avoiding crowded text and arrows in the middle of the graph; instead, highlight the relevant peaks in the graph and report their assignment in a separated table. (ii) Why does the baseline have background at 0.8 in the case of the film? It should be at 1.0. (iii)  Did you perform baseline subtraction only for the film? Why isn't the spectrum of NPs flat?

- Line 92 "some peaks from Si-Hx bonds are indicators of partial hydrogenation". Provide references in support of this statement.

- In many plots, authors use "a.u." as units for the reported quantity. "a.u." means "astronomical units", so it is wrong: please, use correct units.

- Line 271 "For achieving better homogeneity, the experiments were carried out with bacteria suspended in sterile water". Why does the culture medium reduce homogeneity?

- Lines 257-261. This period is not comprehensible, please rephrase it.

- Fugure 3 (b4-b6). Specify the concentration of NPs corresponding to each image.

- See the highlighted words and sentences and the comments in the attached file for other suggested corrections.

Author Response

Comment 1: Authors do not contextualize adequately their research, since they never mention the existing literature about antibacterial sonodynamic therapy, whose state-of-the-art should be discussed in the introduction (see, for example, 10.1039/D2NR01847K and 10.1021/acsbiomaterials.1c00587).

Answer: To contextualize adequately our work we have added several links to papers concerning antibacterial sonodynamic therapy (including the suggested ones) and put some important data about state-of-the-art in the field of research to the Introduction section. The corresponding Refs.[30, 31] as well new Refs. [35-37] have been added and discussed in the revised Manuscript (lines 66-91).

Comment 2: Based on the existing literature, the novelty of this research need to be clarified in the introduction and the new achievements should be highlighted in the conclusions.

Answer: The novelty of research has been additionally clarified in Introduction section of the revised Manuscript. In particular, it has been added that the existing works did not clarify possible impacts of ultra-sound-induced heating and cavitation sensitized by NPs for their antibacterial properties. The corresponding sentences are added in lines 85-97.

Comment 3: In the paragraph of the introduction about the use of PSi NPs for sonodynamic therapy, authors only cite their own work (5 references!): this is inappropriate self-citation. Instead, it would be appropriate to replace most self-citations by other relevant references (as, for example, 10.1186/1556-276X-9-463).

Answer: We reduced the number of self-citations and added other relevant references concerning sonosensitizers for the SDT, combined utilization of nanoparticles and ultrasound, as well as usage of nanoparticles as antibacterial agents. By the way, the Reviewer’s recommended reference (doi: 10.1186/1556-276X-9-463) is the work carried out by the members of our scientific group and collaborators. Therefore, this paper (new Ref.[25]) together with a lot of new references of other researchers [26-31, 35] were additionally cited and discussed in the revised Manuscript.

Comment 4: TEM and DLS techniques are described poorly: all experimental details should be reported accurately.

Answer: Details on the utilized TEM and DLS techniques as well added XRD measurements have added to Section 3.2 (lines 347-363).

Comment 5: Authors should clarify what do they mean with particle size:

  • in TEM, they do not specify how it is evaluated from images;
  • DLS usually measures hydrodynamic radius or hydrodynamic diameter, which one of these quantities is reported in Figure 1?
  •  in general, authors should precisely report the specific measured quantity instead of using the generic word "size".

Answer: We have clarified that the maximal longitudinal dimension of irregular shaped nanoparticles is what is meant by “particle size” for the TEM measurements, however, we left short variant “size” further in the text. References to previously published works (https://doi.org/10.1016/j.micromeso.2021.111641;https://doi.org/10.1016/j.micromeso.2021.111473) where size distributions of similar mesoporous silicon nanoparticles were plotted from their SEM images using neural network algorithms and AFM images were inserted into Section 2.1. Characterization of m-PSi NPs. The DLS technique provided hydrodynamic diameter distributions in the form of a fraction in a total number of particles, so we made the necessary changes across the text, including the axis name in Figure 1c.

Comment 6: Lines 66-70. This period is wrong, since (i) no experiments about colloidal stability in time are reported in the manuscript, and (ii) the Z-potential does not guarantee the stability of a colloidal suspension, as the presence of inhomogeneous charge could induce aggregation even for high Z-potential (see, for example, 10.3390/nano12091529 and 10.1016/j.jcis.2020.07.006): modify the sentences to make them more accurate or remove them.

Answer: We have added DLS data concerning time changes in the hydrodynamic diameter distribution of m-PSi NPs’ suspensions into Figure 1c and came to a conclusion about their short-term colloidal stability. The measured value of zeta-potential (–20 mV) is typical for the investigated m-PSi NPs and corresponds to those obtained previously by other researchers. Absence of any reported data on the instability of suspensions of such kind and any visual sedimentation within several days of storage at room temperature let us suppose that the stability of suspensions is rather determined by their intrinsic properties, i.e., low density and hydrophobicity of as-prepared material, than domination of electrostatic forces over the gravitation ones like, for example, in case of Au or Cd nanoparticles. The corresponding discussion is given in Section 2.1. Characterization of m-PSi NPs (lines 124-139).

Comment 7: Some peaks of the FTIR spectrum that are pointed in Figure 1d are not discussed in the text (Si-O bending, C-H stretching, O-H stretching).

Answer: We gave information about the missed bands in the FTIR spectra in Section 2.1. Characterization of m-PSi NPs, including C-H stretching vibrations attributed to contamination by organic molecules in the process of m-PSi NPs’ synthesis and O-H stretching vibrations from residual water molecules in dried particle powders (lines 165-172).

Comment 8: Line 95. It is not clear how the hydrophilicity of PSi NPs influences their colloidal stability: motivate this statement or remove it.

Answer: We have removed this statement from the text to avoid any misunderstanding.

Comment 9: The quantity subharmonic magnitude should be defined to make the results comprehensible.

Answer: We presented raw spectra (in dB) detected by the hydrophone in the cuvette filled with an investigated sample where either no or some cavitation process takes place in Figure S4 of Supplementary Materials. The position of subharmonic is indicated there. We also replaced Figure 2b by smoothed time curves of subharmonic magnitudes in dB within time domain limited to 60 s of record.

Comment 10: Line 119 "correlates with the more intense cavitation process in the suspension m-PSi NPs". No experiments about the intensity of the cavitation process are reported in the manuscript to substantiate this statement. Remove this sentence.

Answer: It was removed in the revised Manuscript.

Comment 11: Lines 120-121 "This fact can be explained by the greater number of air bubble seeds on the porous surface of m-PSi NPs". No experiments to evaluate the number of bubbles are reported in the manuscript to substantiate this statement. If it is an hypothesis, be clear, otherwise remove the sentence.

Answer: This is our hypothesis; we explained the idea of bubble growths from nanosized air nuclei in particle pores due to the processes of rectified diffusion and bubble coalescence in Section 2.2. Cavitation activity in suspensions of m-PSi NPs.

Comment 12: Figure 2c. Authors state in the text that "The smooth curve is a result of fitting": the function used for the fitting procedure, the motivation for the choice (did you fit experimental data to a theoretical model?), and the fitting parameters must be reported in the manuscript.

Answer: This Bezier spline was utilized just for the convenience of perception. No theoretical model was used.

Comment 13: Figure 2d. The huge increase of the cavitation energy observed at 3.5 W/cm2 in the case c =1 g/L, that is absent in trends at different concentrations, should be discussed in the text of the manuscript.

Answer: We discussed this dramatic increase followed by a steep drop in the subharmonic magnitude at high ultrasound intensities in Section 2.2. Cavitation activity in suspensions of m-PSi NPs and referenced to our previous paper, where bubble dynamics simulation was performed (Tamarov, K.; et. al. Nano air seeds trapped in mesoporous Janus nanoparticles facilitate cavitation and enhance ultrasound imaging. ACS Appl. Mater. Interfaces 2017, 9, 35234–35243, https://doi.org/10.1021/acsami.7b11007).

Comment 14: To substantiate the claims reported in lines 163-167, authors must report experiments (FTIR and cavitation activity) as a function of the aging of NPs.

Answer: We provided both the FTIR and cavitation activity data in Figure 1d and Figure 2b for the freshly prepared and aged suspensions of m-PSi NPs with the relevant discussion of the observed effect in the revised Manuscript (lines 219-277).

Comment 15:

In figure 3a, the following control samples are missing:

- bacteria without application of ultrasound, kept in the same conditions (transfer in the glass tube, thermostatic bath, temperature) and for the same times used in treatments;

- treatments with PSi NPs without application of ultrasound (in this respect, the sentence "m-PSi NPs themselves do not influence the viability of bacteria", reported in lines 176-177, is not supported by experiments).

Answer: We inserted the viability assay in Figure S6 of the Supplementary Materials, as well as discussed the obtained low cytotoxicity of m-PSi NPs in respect to Lactobacillus casei in Section 2.3. Antibacterial effect of ultrasound in the presence of m-PSi NPs.

Comment 16: The deformation/poration of bacterial cells observed by SEM and the (possible, control sample is missing) effects of ultrasound on the bacteria viability should be discussed in terms of ultrasound-induced mechanical and biological damage and compared to other studies performed on this phenomena (see, for example, 10.1038/s41598-017-16708-4)

Answer: We added the discussion of potential mechanical effects of the ultrasound on the bacterial cell membranes and cited the above work in Section 2.3. Antibacterial effect of ultrasound in the presence of m-PSi NPs.

Comment 17: Viability results at the same NPs concentrations used in cavitation experiments must be reported in Figure 3 and, vice versa, cavitation results at the same concentrations used in viability experiments should be reported in Figure 2.

Answer: We understood this comment as a demand for the viability assay. It is addressed above in Comment 15. The concentrations verified in the both cavitation activity measurements and antibacterial experiments were of the same range.

Comment 18: Why did authors performed viability experiments at 1.5 W/cm2 instead of an higher intensity, above the cavitation threshold?

Answer: We stated in the text, that ultrasound intensity of 1.5 W/cm2 was chosen in order to exclude possible undesirable heating of the irradiated area, which is a factor directly influencing the proliferation rate of bacteria.

Comment 19: In the first sentence of the Introduction, authors only list inorganic NPs, why?

Answer: We have mentioned the investigation of antibacterial properties not only of inorganic, but also organic nanoparticles in the Introduction section of the revised Manuscript (lines 38-39).

Comment 20: In the same sentence, authors only cite one review article published in 2012: a number of more up-to-date papers have been published meanwhile and should be cited.

Answer: We have cited  a number of up-to-date papers (Refs. [1-12] dated from 2012 till 2022) concerning antimicrobial properties of various types of nanoparticles in the Introduction section of the revised Manuscript.

Comment 21: Line 62 "particles with sizes from 30 to 100 nm, which can easily attach or/and penetrate through the cell membrane". Provide references in support of this statement.

Answer: References with results supporting the abilities of mesoporous silicon and silica nanoparticles to associate with and internalize into the cells (Bimbo, L.M.; et. al. Biocompatibility of thermally hydrocarbonized porous silicon nanoparticles and their biodistribution in rats. ACS Nano 2009, 4, 3023–3032. https://doi.org/10.1021/nn901657w; Tolstik, E.; et. al. Studies of silicon nanoparticles uptake and biodegradation in cancer cells by Raman spectroscopy. Nanomedicine: NBM 2016, 12, 1931–1940, http://dx.doi.org/10.1016/j.nano.2016.04.004; Gan, Q.; et. al. Effect of size on the cellular endocytosis and controlled release of mesoporous silica nanoparticles for intracellular delivery. Biomed Microdevices 2012, 14, 259–270, https://doi.org/10.1007/s10544-011-9604-9) were added in the revised Manuscript.

Comment 22: Line 63 "Water stands as a surfactant preventing the formation of large agglomerates". In TEM experiments NPs have been dried and water is not present, therefore the reason for reporting this sentence is not clear. Please, clarify this in the manuscript or remove the sentence.

Answer: We have rearranged this part to make it clear that while the advantage of TEM technique is in the opportunity for researchers to get the detailed morphology of nanoparticles (including their porous structure), size distributions of good quality could be obtained using other types of microscopy and special techniques (corresponding references were put into the text). In fact, there is no such huge agglomerates in aqueous suspension as it could be seen in TEM images of dried m-PSi NPs due to the presence of a dispersant (water).

Comment 23: The information given by electron diffraction pattern is not useful for the scope of the manuscript, this result could be removed.

Answer: We received a request from another Reviewer to include XRD data as well, so the crystalline structure of m-PSi NPs seems to be interesting for readers, and the electron diffraction pattern was kept together with added XRD data in the revised Manuscript.

Comment 24: Figure 1c. Did you really pretend derive a (hydrodynamic radius or diameter?) distribution from DLS measurements, based on 30 counts? Please double check the presentation of this result.

Answer: The name of axis was corrected in Figure 1c. In Section 3.2. Morphology of m-PSi NPs we wrote that the software supplied with the DLS apparatus calculated the ratio of particles of a certain size range in a total number of particles. Number of experimental counts was determined automatically based on the autocorrelation function quality, but the output data was the relative fraction of particles in percent.

Comment 25: Figure 1d. This plot is a messy. (i) Please reorganize it avoiding crowded text and arrows in the middle of the graph; instead, highlight the relevant peaks in the graph and report their assignment in a separated table. (ii) Why does the baseline have background at 0.8 in the case of the film? It should be at 1.0. (iii)  Did you perform baseline subtraction only for the film? Why isn't the spectrum of NPs flat?

Answer: We have reorganized the FTIR spectra in Figure 1d: removed text from the plot, indicated key chemical groups and reported them separately in Table 1. The arbitrary units of transmittance carry no considerable meaning, so we decided to clear away the labels of vertical axis in a way it is done in many other papers. The vertical shift between the spectra is convenient for visual perception. We performed the baseline subtraction for both the samples. The transmittance spectrum of m-PSi NPs is not typically flat (see, for example, Riikonen, J.; et. al. Surface chemistry, reactivity, and pore structure of porous silicon oxidized by various methods. Langmuir 2012, 28, 10573–10583. https://doi.org/10.1021/la301642w). Note, the porous silicon film demonstrates significant reflectance in the IR range, while m-PS NPs’ powders were mixed with KBr to form tablets with no such reflectance, but, once again, we do not pretend to compare the absorbance of the two types of samples or approximate number of bonds on their surface. We added this experimental detail to Section 3.3. FTIR measurement.

Comment 26:vLine 92 "some peaks from Si-Hx bonds are indicators of partial hydrogenation". Provide references in support of this statement.

Answer: Two references (Riikonen, J.; et. al. Surface chemistry, reactivity, and pore structure of porous silicon oxidized by various methods. Langmuir 2012, 28, 10573–10583. https://doi.org/10.1021/la301642w; Ogata, Y.; et. al. Hydrogen in porous silicon: vibrational analysis of SiHx species. J. Phys. Chem. B 1997, 101, 1202–1206. https://doi.org/10.1149/1.2043865) were provided in support of the relation between Si-Hx bonds and partial hydrogenation of m-PSi NPs.

Comment 27: In many plots, authors use "a.u." as units for the reported quantity. "a.u." means "astronomical units", so it is wrong: please, use correct units.

Answer: We have corrected abbreviation “a.u.” for the arbitrary units to commonly accepted “arb. un.” in the revised Manuscript.

Comment 28: Line 271 "For achieving better homogeneity, the experiments were carried out with bacteria suspended in sterile water". Why does the culture medium reduce homogeneity?

Answer: We have changed this ambiguous phrase for more simple ones. We meant that bacteria were suspended, not grown on a substrate, which is more advantageous in case of ultrasonic irradiation with a spatially bound beam.

Comment 29: Lines 257-261. This period is not comprehensible, please rephrase it.

Answer: We have rephrased the above-mentioned part concerning our poor attempt to additionally control the level of cavitation intensity by the level of white noise in Section 3.4. Setup for the cavitation activity detection.

Comment 30: Figure 3 (b4-b6). Specify the concentration of NPs corresponding to each image.

Answer: We have specified the concentration of m-PSi NPs of 1 mg/mL added to the culture of bacteria for their SEM visualization.

Comment 31: See the highlighted words and sentences and the comments in the attached file for other suggested corrections.

Answer: We are grateful to the Reviewer for these comments. Most of the suggested corrections were considered, and the corresponding changes were made in the revised Manuscript.

Round 2

Reviewer 3 Report

The authors addressed satisfactorily the majority of the raised issues. Some remaining comments are reported below.

1. Authors added in Figure 1c the hydrodynamic diameter distributions at different aging times as requested, but they removed the initial distribution (t=0) that was shown in the first version of the manuscript: it should be placed back, together with the other distributions.

2. The Bezier spline used by authors in Figure 2c is not a fitting procedure: please, remove the word “fitting” from the caption and the sentence “The smooth curve is a result of fitting” from line 244.

3. Authors should specify in the caption of Figure 3 that the data represented in panel (a) are expressed as percentage respect to the reference colony (analogously to the caption of Figure S6).

4. The comment (n. 17 of the first round) about the concentrations used in cavitation and viability experiments was not addressed properly: authors should use the same concentrations in the two experiments.

5. Lines 400-402 “Unfortunately, the absorption of sound by the m-PSi 400 NPs suspensions was strong enough to provide appropriate level of sample-to-sample 401 sensitivity.”. This sentence is still not clear, maybe authors mean that the adsorption was not strong enough to provide appropriate sensitivity? Please, correct the sentence.

Author Response

Comment 1: Authors added in Figure 1c the hydrodynamic diameter distributions at different aging times as requested, but they removed the initial distribution (t=0) that was shown in the first version of the manuscript: it should be placed back, together with the other distributions.

Answer: We have corrected Figure 1c by adding the graph for as-prepared suspension of m-PSi NPs and the corresponding sentences were added to the figure caption (lines 114-115) and discussion (lines 120-123) in the revised Manuscript.

Comment 2: The Bezier spline used by authors in Figure 2c is not a fitting procedure: please, remove the word “fitting” from the caption and the sentence “The smooth curve is a result of fitting” from line 244.

Answer: We removed the word “fitting” from the text. The following sentence is used “continuous curves are Bezier splines as guides for eyes” (lines 198-199) in the revised Manuscript.

Comment 3: Authors should specify in the caption of Figure 3 that the data represented in panel (a) are expressed as percentage respect to the reference colony (analogously to the caption of Figure S6).

Answer: We have specified the bacterial viability percentage in Figure 3a (lines 304-306) of the revised Manuscript.

Comment 4: The comment (n. 17 of the first round) about the concentrations used in cavitation and viability experiments was not addressed properly: authors should use the same concentrations in the two experiments.

Answer: We have added concentration dependence for the same US intensity, which was used for the viability experiments, and modified Figure 2c was inserted and briefly discussed to the revised Manuscript (lines 197-198 and 246-248).

Comment 5: Lines 400-402 “Unfortunately, the absorption of sound by the m-PSi 400 NPs suspensions was strong enough to provide appropriate level of sample-to-sample 401 sensitivity.”. This sentence is still not clear, maybe authors mean that the adsorption was not strong enough to provide appropriate sensitivity? Please, correct the sentence.

Answer: The absorption was strong enough to produce different results in white noise spectrum (it was detected with a low sensitivity), so the detection of subharmonic magnitude enables us to overcome this problem and detect the cavitation intensity for the samples.  The corrected sentence was added in the revised Manuscript (lines 402-405).